# DISCOVERING ENVIRONMENTS WITH XRM

## ABSTRACT

Environment annotations are essential for the success of many out-of-distribution generalization methods. Unfortunately, these are resource-intensive to obtain, and their relevance to model performance is limited by the expectations and perceptual biases of human annotators. Therefore, to enable robust AI systems across applications, we must develop algorithms to automatically discover environments inducing broad generalization. Current proposals, which divide examples based on their training error, suffer from one fundamental problem. These methods add hyper-parameters and early-stopping criteria that are impossible to tune without a validation set with human-annotated environments, the very information subject to discovery. In this paper, we propose CROSS-RISK MINIMIZATION (XRM) to address this issue. XRM trains two twin networks, each learning from one random half of the training data, while imitating confident held-out mistakes made by its sibling. XRM provides a recipe for hyper-parameter tuning, does not require early-stopping, and can discover environments for all training and validation data. Domain generalization algorithms built on top of XRM environments achieve oracle worst-group-accuracy, solving a long-standing problem in out-of-distribution generalization.

## 1 INTRODUCTION

AI systems pervade our lives, spanning applications such as finance (Hand and Henley, 1997), healthcare (Jiang et al., 2017), self-driving vehicles (Bojarski et al., 2016), and justice (Angwin et al., 2016). While machines appear to outperform humans on such tasks, these systems fall apart when deployed in testing conditions different to their experienced *training environments* (Geirhos et al., 2020). For instance, during the COVID-19 pandemic, it was shown that thoracic x-ray classifiers latched onto spurious correlations—such as patient age, scanning position, or text fonts—as shortcuts to minimize their own training error (Heaven, 2021). This resulted in "an alarming situation in which the systems appear accurate, but fail when tested in new hospitals" (DeGrave et al., 2021).

Generally speaking, AI systems perform worse on groups of examples under-represented in the training data (Barocas et al., 2019). To drive this point home, consider the left side of figure 1, illustrating the Waterbirds problem (Sagawa et al., 2019). This task considers two class labels, landbirds and waterbirds, collected in two landscape environments, land and water. These combine into four groups: a *majority group* of waterbirds in water (73% of training examples), landbirds in land (22%), waterbirds in land (4%), and a *minority group* of landbirds in water (1%). On this problem, learning machines latch onto the *landscape* spurious feature, because it can separate the majority of examples, exhibits a larger signal-to-noise ratio, and yields the maximum-margin classifier amongst those with zero training error. For instance, consider an empirical risk minimization (Vapnik, 1998, ERM) baseline, which ignores environment information. As shown in the right panel of figure 1, ERM results in a worst-group-accuracy of 61%—attained, in fact, on the minority group.

To improve upon ERM, researchers have developed a myriad of domain generalization (DG) algorithms (Zhou et al., 2022a; Wang et al., 2021). These methods consider environment annotations to uncover invariant (environment-generic) patterns and discard spurious (environment-specific) correlations (Arjovsky et al., 2019). As figure 1 shows, the DG algorithm *group distributionally robust optimization* (Sagawa et al., 2019, GroupDRO) achieves a worst-group-accuracy of 87%. This outperforms ERM by over twenty five points, a sizeable gap!

While promising, DG algorithms require environment annotations. These are resource-intensive to obtain, and their relevance to downstream model performance is limited by the expectations, precision,

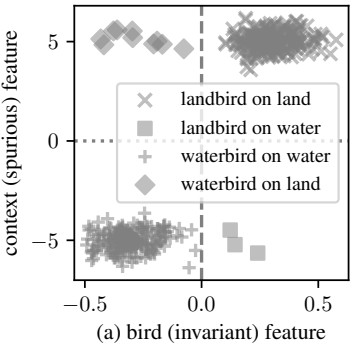 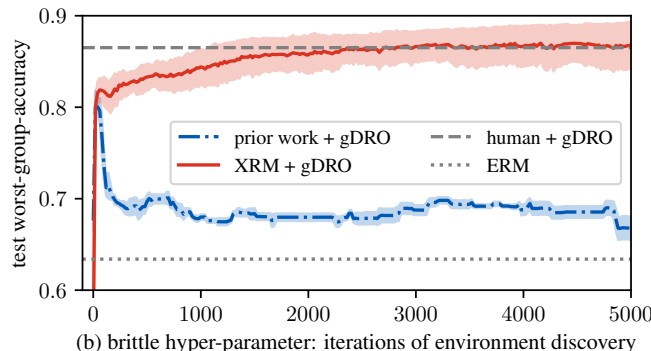

(a) bird (invariant) feature

(b) brittle hyper-parameter: iterations of environment discovery

Figure 1: (a) Waterbirds classification problem, containing four groups: a *majority group* of waterbirds in water, landbirds in land, waterbirds in land, and a *minority group* of landbirds in water. Learning machines latch onto spurious landscape features, as it allows to classify the majority of examples, exhibits a larger signal-to-noise ratio, and yields the maximum-margin classifier amongst solutions with zero training error. (b) Worst-group-accuracy (minority-group-accuracy) for different methods. (Dotted line) An ERM baseline ignoring group annotations achieves 61%. (Dashed line) The GroupDRO domain generalization method with human group annotations, considered our oracle, achieves 87%. (Dashdot blue line) Prior work to discover groups requires early-stopping with surgical precision. This is an impossible ask without a validation set with human group annotations, the very information subject to discovery. Realistically, these methods would converge at 68%. (Solid red line) Our proposed XRM enables an oracle performance of 87% at convergence.

and perceptual biases of human annotators. Moreover, no fixed environment annotations that we may choose and revise can serve the idiosyncrasies of all the different DG algorithms. *In extremis*, and by virtue of the combinatorial-explosion of ways in which two examples can be similar or dissimilar (Goodman, 1972), the patterns that deceive a learning system could be alien or invisible to our human eyes (Goodfellow et al., 2014). Because of these reasons, robust AI systems are currently confined to small data collections, and their promise in the large-scale setting remains unfulfilled.

In light of the above, some researchers began developing algorithms for the automatic discovery of environments from data (Bao and Barzilay, 2022; Zheran Liu et al., 2021; Zhang et al., 2022b; Lahoti et al., 2020; Dagaev et al., 2021; Creager et al., 2020; Nam et al., 2020). In broad strokes, these methods build a robust system in two phases. In phase-1, these methods learn a label predictor to distribute training examples in two environments, based on their training error. In phase-2, a DG algorithm is trained on top of the discovered environments to produce the robust system of interest. Unfortunately, this pipeline suffers from one fundamental issue. In particular, phase-1 needs to control the capacity of the label predictor with surgical precision, such that the discovered environments differ only in spurious correlation. As illustrated in figure 1, these methods discover environments that yield strong phase-2 systems only at the knife's edge of very precisely early-stopped phase-1 training iterations—yielding a decent but still worse than oracle performance of 79%—and otherwise converge to a worst-group-accuracy of 68%. Unfortunately, we lack a signal for such a fine call, so methods for environment discovery resort to a validation set with human environment annotations. In practice, this wraps phase-1 and phase-2 with a cross-validation envelope to directly select a model maximizing validation worst-group-accuracy. Alas, at least in our view, this defeats the *raison d'être* of environment discovery.

*Contribution*   We propose CROSS-RISK MINIMIZATION (XRM), a simple method for environment discovery that requires no human environment annotations whatsoever. XRM trains two twin label predictors, each holding-in one random half of the training data. During training, XRM instructs each twin to imitate confident held-out mistakes made by their sibling. This results in an "echo-chamber" where twins increasingly rely on bias, converging on a pair of environments that differ in spurious correlation, and share the invariances that fuel downstream out-of-distribution generalization. After twin training, a simple cross-mistake formula allows XRM to annotate all of the training and validation examples with environments. As our experiments show, XRM endows DG algorithms with oracle-like performance across benchmarks, solving a long-standing problem in out-of-distribution generalization. Returning one final time to figure 1, we observe that XRM+GroupDRO converges to 87% worst-group-accuracy on Waterbirds, matching the oracle!

The sequel is as follows. Section 2 reviews the setup of DG, pointing environment annotations as one of its major shortcomings. Section 3 surveys prior works on environment discovery, their issues surrounding hyper-parameter tuning and early-stopping, and their consequent reliance on validation sets with human environment annotations. For a more exhaustive review, see section 6. Section 4 introduces XRM, a novel objective to discover machine-tailored environments for all training and validation data. Experiments in Section 5 show that DG algorithms built on top of XRM environments achieve oracle-like performance, and Section 6 closes with thoughts for future work.

## 2   LEARNING INVARIANCES ACROSS KNOWN ENVIRONMENTS

In domain generalization (DG), our goal is to build learning systems that perform well beyond the distribution of the training data. To this end, we collect examples under multiple training environments. Then, DG algorithms search for patterns that are invariant across these training environments—more likely to hold during test time—while discarding environment-specific spurious correlations (Arjovsky et al., 2019). More formally, we would like to learn a predictor $f$ to classify inputs $x$ into their appropriate labels $y$, and across all relevant environments $e \in \mathcal{E}$:

$$f \in \operatorname*{argmin}_{\tilde{f}} \sup_{e \in \mathcal{E}} R^e(\tilde{f}), \tag{1}$$

where the risk $R^e(f) = \mathbb{E}_{(x,y) \sim P^e}[\ell(f(x), y)]$ measures the average loss $\ell$ incurred by the predictor $f$ across examples from environment $e$, all of them drawn iid from $P^e$.

In practical applications, the DG problem (1) is under-specified in two important ways. Firstly, we only get to train on a subset of all of the relevant environments $\mathcal{E}$, called the training environments $\mathcal{E}_{\mathrm{tr}} \subset \mathcal{E}$. Yet, the quality of our predictor continues to be the worst classification accuracy across *all* environments $\mathcal{E}$. Secondly, and for each training environment $e \in \mathcal{E}_{\mathrm{tr}}$, we do not observe its entire data distribution $P^e$, but only a finite set of iid examples $(x_i, y_i, e_i = e)$. In sum, the information at our disposal to address the DG problem (1) is the training dataset $\mathcal{D}_{\mathrm{tr}} = \{(x_i, y_i, e_i)\}_{i=1}^n$, where $(x_i, y_i)$ is an input-label pair drawn from the distribution $P^{e_i}$ associated to the training environment $e_i \in \mathcal{E}_{\mathrm{tr}}$. Armed with $\mathcal{D}_{\mathrm{tr}}$, we approximate (1) by the observable optimization problem

$$f \in \operatorname*{argmin}_{\tilde{f}} \sup_{e \in \mathcal{E}_{\mathrm{tr}}} R_n^e(\tilde{f}), \tag{2}$$

where $R_n^e(f) = \frac{1}{|\mathcal{D}_{\mathrm{tr}}^e|} \sum_{(x_i, y_i) \in \mathcal{D}_{\mathrm{tr}}^e} \ell(f(x_i), y_i)$ is the empirical risk (Vapnik, 1998) across the data $\mathcal{D}_{\mathrm{tr}}^e = \{(x_i, y_i) \in \mathcal{D}_{\mathrm{tr}} : e_i = e\}$ from the training environment $e \in \mathcal{E}_{\mathrm{tr}}$.

*Environments and groups*   In its full generality, domain generalization is an admittedly daunting task. To alleviate the burden, much prior literature considers the simplified version of *group shift* (Sagawa et al., 2019). The problem formulation is equivalent: we observe each input $x_i$ together with some attribute $e_i$ and label $y_i$, and define one *group* $g \equiv e \times y$ per attribute-label combination. (We use the terms "environment" and "attribute" interchangeably.) Again, we assume access to one training set of triplets $(x_i, y_i, e_i)$ to learn our predictor, and one similarly formatted validation set available for hyper-parameter tuning and model selection purposes. Next, we put in place one important simplifying assumption $\mathcal{E} = \mathcal{E}_{\mathrm{tr}}$, namely no new environments appear during test time. Consequently, the quality of our predictor can be directly estimated as the worst-group-accuracy in the validation set. Because most learning algorithms focus on minimizing *average* training error, oftentimes the worst-group-accuracy happens to be the accuracy at the *minority group*.

In practice, different DG algorithms (Gulrajani and Lopez-Paz, 2020; Zhou et al., 2022a; Wang et al., 2021; Yang et al., 2023) target different types of invariance, learned across training environments $\mathcal{E}_{\mathrm{tr}}$, assumed to hold across testing environments $\mathcal{E}_{\mathrm{te}}$, and implemented as various innovations to the objective (2). As discussed in section 1, some DG algorithms outperform by a large margin methods ignoring environment (*cf.* attribute, group) information, such as ERM.

Despite their promise, the main roadblock towards large-scale domain generalization is their reliance on humanly annotated environments, attributes, or groups. These annotations are *resource-intensive* to obtain. Moreover, the expectations, precision, and perceptual biases of annotators can lead to environments conducive of sub-optimal out-of-distribution generalization. Different machine learning models fall prey to different kinds of spurious correlations. In addition, there are plenty

of complex and subtle interactions between environment definition, function class, distributional shift, and cultural viewpoint (Lopez-Paz et al., 2022). Therefore, environment annotations are helpful only when revealing spurious and invariant patterns under the lens of the learning system under consideration. Could it be possible to design algorithms for the automatic discovery of environments tailored to the learning machine and data at hand?

## 3 DISCOVERING ENVIRONMENTS
*Nature does not shuffle data—Bottou (2019)*

Let us reconsider the problem of domain generalization without access to environment annotations. This time it suffices to talk about one training distribution $P^{\text{tr}}$ and one testing distribution $P^{\text{te}}$. Our training data is a collection of input-label pairs $(x_i, y_i)$, each drawn iid from the training distribution. While the training distribution $P^{\text{tr}}$ may be the mixture of multiple environments describing interesting invariant and spurious correlations, this rich heterogeneity is shuffled together and unbeknown to us. But, if we could "unshuffle" the training distribution and recover the environments therein, we could invoke the domain generalization machinery from the previous section and hope for a robust predictor. This is the purpose of automatic environment discovery.

To discover environments and learn from them, most prior work implements a pipeline with two phases. On phase-1, train a label predictor and distribute each training example into two environments, depending on whether the example is correctly or incorrectly classified. On phase-2, train a DG algorithm on top of the discovered environments. Crucially, one must control the capacity of the label predictor in phase-1 with surgical precision, such that it relies only on prominent, easier-to-learn spurious correlations. If the environments discovered in phase-1 differ only in spurious correlation, as we would like, then the DG algorithm from phase-2 should be able to zero-in on invariant patterns more likely to generalize to the test distribution $P^{\text{te}}$. On the unlucky side, if phase-1 produces a zero-training-error predictor, we would be providing phase-2 with one non-vacuous, non-informative environment—the training data itself!

As a result, proposals for environment discovery differ mainly in how to control the capacity of the phase-1 label predictor. For example, the too-good-to-be-true prior (Dagaev et al., 2021) employs a predictor with a small parameter count while correct-n-contrast (Zhang et al., 2022b, CnC) applies strong weight decay regularization. Just train twice (Zheran Liu et al., 2021, JTT) and environment inference for invariant learning (Creager et al., 2020, EIIL) train a phase-1 predictor for a limited number of epochs. Learning from failure (Nam et al., 2020, LfF) biases the predictor towards the use of "simple" features by applying a generalized version of the cross entropy loss. Other proposals, such as learning to split (Bao and Barzilay, 2022, LS) and adversarial re-weighted learning (Lahoti et al., 2020, ARL) complement capacity control with adversarial games.

However regularized, all of these methods suffer from one fundamental problem. More specifically, these phase-1 strategies add hyper-parameters and early-stopping criteria, but remain silent on how to tune them. As illustrated in figure 1 for Waterbirds, methods like the above discover environments leading to competitive generalization only when phase-1 is trained for a number of iterations that fall within a knife's edge. Quickly after that, the performance of the resulting phase-2 system falls off a cliff, landing at ERM-like worst-group-accuracy.

Prior works keep away from this predicament by assuming a validation set with human environment annotations. Then, it becomes possible to simply wrap phase-1 and phase-2 into a cross-validation pipeline that promotes validation worst-group-accuracy. Alas, this defeats the entire purpose of environment discovery. In fact, if we have access to a small dataset with human environment annotations, these examples suffice to fine-tune the last layer of a deep neural network towards state-of-the-art worst-group-accuracy (Izmailov et al., 2022). Looking forward, could we develop an algorithm for environment discovery that requires no human annotations whatsoever, and robustly yields oracle-like phase-2 performance?

## 4 CROSS-RISK MINIMIZATION (XRM)

We propose CROSS-RISK MINIMIZATION (XRM), an algorithm to discover environments without the need of human supervision. XRM comes with batteries included, namely a recipe for hyper-parameter tuning and a formula to annotate all training and validation data. As we will show in section 5, environments discovered by XRM endow phase-2 DG algorithms with oracle performance.

The blueprint for phase-1 with XRM is as follows. XRM trains two twin label predictors, each holding-in one random half of the training data (section 4.1). During training, XRM biases each twin to absorb spurious correlation by imitating confident held-out mistakes from their sibling (section 4.2). XRM chooses hyper-parameters for the twins based on the number of imitated mistakes (section 4.3). Finally, and given the selected twins, XRM employs a simple "cross-mistake" formula to discover environment annotations for all of the training and validation examples (section 4.4). Algorithm 1 serves as a companion to the descriptions below; appendix G contains a real PyTorch implementation. The runtime of phase-1 with XRM is akin to one ERM baseline on the training data.

---

**Algorithm 1** CROSS-RISK MINIMIZATION (XRM)

**Input:** training examples $\{(x_i, y_i)\}_{i=1}^n$ and validation examples $\{(\tilde{x}_i, \tilde{y}_i)\}_{i=1}^m$
**Output:** discovered environments for training $\{e_i\}_{i=1}^n$ and validation $\{\tilde{e}_i\}_{i=1}^m$ examples

- Fix held-in training example assignments $m_i^a \sim \text{Bernoulli}(\frac{1}{2})$ and $m_i^b = 1 - m_i^a$
- Init two label predictors $f^a$ and $f^b$ at random; calibrate softmax temperatures on held-in data
- Until convergence:
    - Compute held-in softmax predictions $p_i^{\text{in}} = m_i^a f^a(x_i) + m_i^b f^b(x_i)$
    - Compute held-out softmax predictions $p_i^{\text{out}} = m_i^b f^a(x_i) + m_i^a f^b(x_i)$
    - Update $f^a$ and $f^b$ to minimize the class-balanced held-in cross-entropy loss $\ell(p^{\text{in}}, y)$
    - Flip $y_i$ into $y_i^{\text{out}} = \text{argmax}_j p_{i,j}^{\text{out}}$, with prob. $(p_{i,y_i^{\text{out}}}^{\text{out}} - 1/n_{\text{classes}}) \cdot n_{\text{classes}}/(n_{\text{classes}} - 1)$
- Define cross-mistake function $e(x, y) = [\![(y \notin \text{argmax}_j f^a(x)_j) \vee (y \notin \text{argmax}_j f^b(x)_j)]\!]$
- Discover training $e_i = e(x_i, y_i)$ and validation $\tilde{e}_i = e(\tilde{x}_i, \tilde{y}_i)$ environments

---

## 4.1 TWIN SETUP, HOLDING-OUT OF DATA

We start by initializing two twin label predictors $f^a$ and $f^b$. Without loss of generality, let these predictors return softmax probability vectors over the $n_{\text{classes}}$ classes in the training data. We split our training dataset $\{(x_i, y_i)\}_{i=1}^n$ in two random halves. Formally, we construct a pair of training assignment vectors with entries $m_i^a \sim \text{Bernoulli}(\frac{1}{2})$ and $m_i^b = 1 - m_i^a$, for all $i = 1, \ldots, n$. For predictor $f^a$, examples with $m_i^a = 1$ are "held-in" and examples with $m_i^a = 0$ are "held-out"; similarly for $f^b$. Therefore, we will train predictor $f^a$ on training examples where $m_i^a = 1$, and similarly for predictor $f^b$. Before learning starts, we calibrate the softmax temperature of the twins via Platt scaling (Guo et al., 2017). See appendix G for implementation details.

By virtue of this arrangement, we may now estimate the generalization difficulty of any example by looking at the prediction of the twin that held-out such point. This contrasts prior methods for environment discovery, which consume the entire training data, and may therefore conflate generalization and memorization. Here, however, if a point is misclassified when held-out, we see this as evidence of such example belonging to the minority group. Feldman and Zhang (2020) proposes a similar "error when holding-out" construction as a measure of memorization. CrossSplit (Kim et al., 2023) also employs this mechanism to avoid memorization of noisy labels in the context of label-noise robustness. As a last remark, we recommend choosing the twins to inhabit the same function class as the downstream phase-2 DG predictor, to discover environments tailored to the paticular learning machine.

## 4.2 TWIN TRAINING, FLIPPING LABELS

As figure 1 shows, the test worst-group-accuracy of an ERM baseline on Waterbirds is 62%. This suggests that, if using ERM to train our twins, each would be able to correctly classify roughly one half of the minority examples. If using these machines to discover environments based on prediction errors, we would dilute the spurious correlation evenly across the two discovered environments. Consequently, it would be difficult for a phase-2 DG algorithm to tell apart between invariant and spurious patterns. Albeit counter-intuitive, we would like to hinder the learning process of our twins, such that they increasingly rely on spurious correlation. In the best possible case, the twins

would correctly classify all majority examples and mistake all minority examples, resulting in *zero* worst-group accuracy.

To this end, we propose to steer away our twins from becoming empirical risk minimizers as follows. Let $p_i^{\text{out}} = m_i^b f^a(x_i) + m_i^a f^b(x_i)$ be the held-out softmax prediction for example $(x_i, y_i)$. Also, let $y_i^{\text{out}} = \arg\max_j p_{i,j}^{\text{out}}$ be the held-out predicted class label, equal to the index of the maximum held-out softmax prediction. Then, at each iteration during the training of the twins,

$$\text{flip } y_i \text{ into } y_i^{\text{out}}, \text{ with probability } (p_{y_i^{\text{out}}}^{\text{out}} - 1/n_{\text{classes}}) \cdot n_{\text{classes}}/(n_{\text{classes}} - 1), \tag{3}$$

and let each twin take a gradient step to minimize their held-in class-balanced—according to the moving targets—cross-entropy loss.

The overarching intuition is that the label flipping equation (3) implements an "echo chamber" reinforcing the twins to rely on spurious correlation. Label flipping happens more often for confident held-out mistakes and early in training. These are two footprints of spurious correlations, since these are often easier and faster to capture. (In the context of neural networks, this is often referred to as a "simplicity bias" (Shah et al., 2020b; Pezeshki et al., 2021).) Overall, the purpose of equation (3) is to transform the labels of the training data such that they do not longer represent the original classes, but spurious bias. Finally, the adjustment of equation (3) in terms of $n_{\text{classes}}$ ensures low flip probabilities at initialization, where most mistakes are due to weight randomness, and not due to spurious correlation. The aligning "echo chamber" effect from label-flipping is a crucial novelty compared to methods that use multiple networks to either disagree with or diversify spurious features. (Nam et al., 2020; Cha et al., 2021; Rame et al., 2022; Wortsman et al., 2022; Lee et al., 2023; Pagliardini et al., 2023; Lin et al., 2023; Eastwood et al., 2023).

### 4.3 TWIN MODEL SELECTION, COUNTING LABEL FLIPS

Before discovering environments, we must commit to a pair of twin predictors. Since these have their own hyper-parameters, XRM would be incomplete without a phase-1 model selection criterion (Gulrajani and Lopez-Paz, 2020). We propose to select the twin hyper-parameters showing a maximum number of label flips (3) at the last iteration, and across the training data. To reiterate, by "counting flips" we simply compare the vector of current labels with the vector of original labels—therefore, we do not accumulate counts of double or multiple flips per label. To understand why, recall that each label flip signifies one example that is confidently misclassified when held-out. Therefore, each label flip is evidence about reliance on spurious correlation, which consequently brings us closer to a clear-cut identification of the minority group.

### 4.4 ENVIRONMENT DISCOVERY, USING CROSS-MISTAKE FORMULA

Having committed to a pair of twins, we are ready to discover environments for all of our training and validation examples. In particular, we use a simple "cross-mistake" formula to annotate any example $(x, y)$ with the binary environment

$$e(x, y) = [\![ (y \notin \text{argmax}_j f^a(x)_j) \lor (y \notin \text{argmax}_j f^b(x)_j) ]\!]. \tag{4}$$

where "$\lor$" denotes logical-OR, and "$[\![ \, ]\!]$" is the Iverson bracket. If operating within the group-shift paradigm, finish by defining one group per combination of label and discovered environment. Notably, the ability to annotate both training and validation examples is a feature inherited from holding-out data during twin training. More particularly, every example—within training and validation sets—is held-out for at least one of the two twins, as subsumed in equation (4) by the logical-OR operation.

While (4) partitions data into "only" two environments, we do not believe that XRM is identifying "one spurious correlation", but a generic "direction of spuriousness". We suspect that an Oracle with perfect knowledge about the test domain would be able to split the training data in two environments that maximally reveal the train-test drift. Furthermore, alternative equations to (4) could quantize continuous loss values into multiple bins to generate many environments with XRM.

We are now ready to train the phase-2 DG algorithm of our choice on top of the training data with environments discovered with XRM. When doing so, we can perform phase-2 DG model selection by maximizing worst-group-accuracy on the validation data with environments discovered by XRM.

## 5 EXPERIMENTS

Our experimental protocol has three moving pieces: datasets, phase-2 domain generalization algorithms, and the source of environment annotations. For a tour on various versions of the ColoredMNIST benchmark providing intuitions on when XRM may work or fail, please refer to section 6. For results on the popular DomainBed benchmark, please refer to section 6.

*Datasets*   We consider six standard datasets from the SubpopBench suite (Yang et al., 2023). These are the four image datasets Waterbirds (Wah et al., 2011), CelebA (Liu et al., 2015), MetaShift (Liang and Zou, 2022), and ImageNetBG (Xiao et al., 2020); and the two natural language datasets MultiNLI (Williams et al., 2017) and CivilComments (Borkan et al., 2019). We invite the reader to Appendix B.1 from Yang et al. (2023) for a detailed description of these. For CelebA, predictors map pixel intensities into a binary "blonde/not-blonde" label. No individual face characteristics, landmarks, keypoints, facial mapping, metadata, or any other information was used to train our CelebA predictors. We also conduct experiments on ColorMNIST (Arjovsky et al., 2019), but keep a strict protocol. More specifically, we set *both* training and validation data to contain two environments, with 0.8 and 0.9 label-color correlation, while the test environment shows 0.1 label-color correlation. This contrasts Arjovsky et al. (2019), who used the test environment for model selection.

*Phase-2 DG algorithms*   We consider ERM, group distributionally robust optimization (Sagawa et al., 2019, GroupDRO), group re-weighting (Japkowicz, 2000, RWG), and group sub-sampling (Idrissi et al., 2022, SUBG). When group information is available, we tune hyper-parameters and early-stopping by maximizing worst-group-accuracy. Otherwise, we tune for worst-class-accuracy. Following standard praxis, image datasets employ a pretrained ResNet-50, while text datasets use a pretrained BERT. For more details, see appendix D.

*Environment annotations*   For each combination of dataset and phase-2 DG algorithm, we compare group annotations from different sources. *None* denotes no group annotations. *Human* denotes ground-truth annotations, as originally provided in the datasets, and inducing oracle performance. *XRM* denotes group annotations from the environments discovered by our proposed method. In some experiments we compare XRM to other environment discovery methods, these being learning from failure (Nam et al., 2020, LfF), environment inference for invariant learning (Creager et al., 2020, EIIL), just train twice (Zheran Liu et al., 2021, JTT), correct-n-contrast (Zhang et al., 2022b, CnC), automatic feature re-weighting (Qiu et al., 2023, AFR), and LS (Bao and Barzilay, 2022).

*Metrics*   Regardless of how training and validation groups are discovered, we always report test worst-group-accuracy over the human group annotations provided by each dataset. The tables hereby presented show averages over ten random seeds. For tables with error bars, see appendix F.

### 5.1 XRM VERSUS HUMAN ANNOTATIONS

Table 1 shows that XRM enables oracle-like worst-group-accuracy across datasets. The performance gains are remarkable in the challenging ColorMNIST dataset, where XRM perfectly identifies digits appearing in minority colors, discovering a pair of environments conducive of stronger generalization than the ones originally proposed by humans. For the commonly-reported quartet of Waterbirds, CelebA, MultiNLI, and CivilComments, human annotations induce an average oracle worst-group-accuracy 80.6%, while XRM environments endow a super-human performance of 80.9%.

### 5.2 XRM VERSUS OTHER METHODS FOR ENVIRONMENT DISCOVERY

Table 2 shows the worst-group-accuracy of GroupDRO when built on top of environments as discovered by different methods. As seen in the previous subsection, XRM achieves 80.4%, nearly matching oracle performance. The second best method with no access to environment information, JTT, drops to 58.9%. The best method accessing a validation set with human environment annotations, AFR, lags far from XRM, with 78%. The computational burden to complete the results from LS was prohibitive. For example, one run of LS for Waterbirds, the smallest dataset, took 20 hours. An XRM run for this same dataset, on the same 32GB Volta GPU, takes 10 minutes. As detailed

Table 1: Worst-group-accuracies across datasets and algorithms average over ten random runs, with XRM showing oracle-level results. Remark 1: Class labels substitute group labels when the latter are not available. Remark 2: ERM, while not trained with group labels, can still benefit from validation group labels for hyperparameter tuning, leading to better performance.

|  | ERM | | | GroupDRO | | | RWG | | | SUBG | | |
|---|---|---|---|---|---|---|---|---|---|---|---|---|
|  | None | Human | **XRM** | None | Human | **XRM** | None | Human | **XRM** | None | Human | **XRM** |
| **Waterbirds** | 70.4 | 76.1 | 75.3 | 71.7 | 88.0 | 86.1 | 74.8 | 87.0 | 84.5 | 73.0 | 86.7 | 76.3 |
| **CelebA** | 62.7 | 71.9 | 67.6 | 68.8 | 89.1 | 89.8 | 70.7 | 89.5 | 88.0 | 68.5 | 87.1 | 87.5 |
| **MultiNLI** | 54.8 | 65.3 | 65.8 | 69.7 | 76.1 | 74.3 | 69.3 | 71.1 | 73.3 | 53.7 | 72.8 | 71.3 |
| **CivilComments** | 55.1 | 59.7 | 61.4 | 59.3 | 69.3 | 71.3 | 54.0 | 65.8 | 73.4 | 58.7 | 64.0 | 72.9 |
| **ColorMNIST** | 10.1 | 10.1 | 26.9 | 10.0 | 10.1 | 70.5 | 10.1 | 10.2 | 70.2 | 10.1 | 10.0 | 70.8 |
| **MetaShift** | 70.8 | 67.7 | 71.2 | 70.8 | 73.7 | 71.2 | 66.5 | 70.8 | 69.7 | 70.0 | 73.5 | 69.2 |
| **ImagenetBG** | 75.6 | 76.9 | 76.9 | 73.0 | 76.1 | 75.9 | 76.9 | 76.5 | 77.0 | 75.0 | 75.0 | 76.4 |
| **Average** | 57.1 | 61.1 | 63.6 | 60.5 | 68.9 | 77.0 | 60.3 | 67.3 | 76.6 | 58.4 | 67.0 | 74.9 |

Table 2: Average/worst accuracies comparing methods for environment discovery. We specify access to annotations in training data ($e^{tr}$) and validation data ($e^{va}$). Symbol † denotes original numbers.

| $e^{tr}$ | $e^{va}$ |  | Waterbirds | | CelebA | | MNLI | | CivilComments | | Average | |
|---|---|---|---|---|---|---|---|---|---|---|---|---|
|  |  |  | Avg | Worst | Avg | Worst | Avg | Worst | Avg | Worst | Avg | Worst |
| ✓ | ✓ | ERM | 86.1 | 76.1 | 93.5 | 71.9 | 78.6 | 65.3 | 82.9 | 59.7 | 85.3 | 68.3 |
|  |  | GroupDRO | 92.6 | 88.0 | 93.3 | 89.1 | 82.0 | 76.1 | 81.4 | 69.3 | 87.3 | 80.6 |
| ✗ | ✓ | ERM† | 97.3 | 72.6 | 95.6 | 47.2 | 82.4 | 67.9 | 83.1 | 69.5 | 89.6 | 64.3 |
|  |  | LfF† | 91.2 | 78.0 | 85.1 | 77.2 | 80.8 | 70.2 | 68.2 | 50.3 | 81.3 | 68.9 |
|  |  | EIIL† | 96.9 | 78.7 | 89.5 | 77.8 | 79.4 | 70.0 | 90.5 | 67.0 | 89.1 | 73.4 |
|  |  | JTT† | 93.3 | 86.7 | 88.0 | 81.1 | 78.6 | 72.6 | 83.3 | 64.3 | 85.8 | 76.2 |
|  |  | CnC† | 90.9 | 88.5 | 89.9 | 88.8 | — | — | — | — | — | — |
|  |  | AFR† | 94.4 | 90.4 | 91.3 | 82.0 | 81.4 | 73.4 | 89.8 | 68.7 | 89.2 | 78.6 |
| ✗ | ✗ | ERM | 85.3 | 70.4 | 94.5 | 62.7 | 77.9 | 54.8 | 80.9 | 55.1 | 84.6 | 60.8 |
|  |  | LfF† | 86.6 | 75.0 | 81.1 | 53.0 | 71.4 | 57.3 | 69.1 | 42.2 | 77.1 | 56.9 |
|  |  | EIIL† | 90.8 | 64.5 | 95.7 | 41.7 | 80.3 | 64.7 | — | — | — | — |
|  |  | JTT† | 88.9 | 71.2 | 95.9 | 48.3 | 81.4 | 65.1 | 79.0 | 51.0 | 86.3 | 58.9 |
|  |  | LS† | 91.2 | 86.1 | 87.2 | 83.3 | 78.7 | 72.1 | — | — | — | — |
|  |  | BAM† | 91.4 | 89.1 | 88.4 | 80.1 | 80.3 | 70.8 | 88.3 | 79.3 | 87.1 | 79.8 |
|  |  | **XRM** | 90.6 | 86.1 | 91.8 | 91.8 | 78.3 | 74.3 | 79.9 | 71.3 | 85.2 | 80.9 |

in appendix E, we found that by varying some of the fixed hyper-parameters in the official LS repository, performance on Waterbirds varied by as much as $\pm 7\%$ in worst-group accuracy.

## 5.3 SOME VISUALIZATIONS

Figure 2 explores some of the behaviors of XRM on the Waterbirds dataset. In particular, the left panel justifies the use of "percentage of label flipped at convergence" as a phase-1 model selection criterion for XRM, as it correlates strongly with downstream phase-2 worst-group-accuracy. The two middle panels showcase the clear separation of the minority group "landbirds/water" by XRM, as no landbirds in land are in the cross-mistake area. The right panel shows that label flipping happens almost exclusively for minority groups, and converges alongside XRM training. This provides XRM with a degree of stability, removing the need for intricate early-stopping criteria.

Figure 3 applies XRM to the CIFAR-10 dataset (Krizhevsky et al., 2009). While CIFAR-10 does not contain environment annotations, the discovered environments by XRM for the "plane" and "deer" classes reveal one interesting spurious correlations, namely background color. As a final remark, we ablated the need for (i) holding-out data, and (ii) performing label flipping, finding that both components are essential to the performance of XRM.

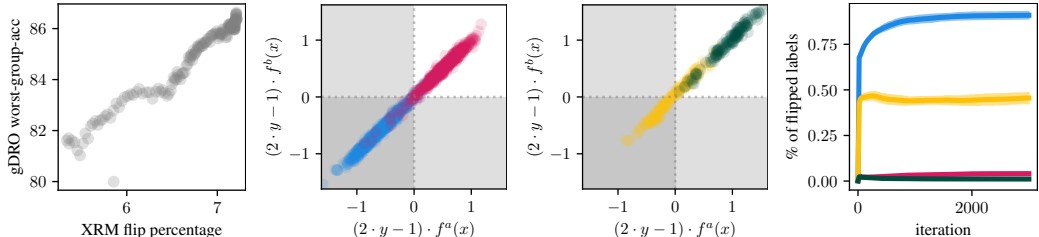

Figure 2: XRM on the Waterbirds problem, concerning ■ waterbirds in water, ■ waterbirds in land, ■ landbirds in water, ■ landbirds in land. The first panel shows that "percentage of XRM label flipped at convergence" is a strong indicator of "worst-group-accuracy in phase-2", making flips a good criterion to select twin hyper-parameters. The two middle panels show the signed margin of the twins on each ground-truth group. From each of these class-dependent plots, XRM discovers two environments: one for points in the "mistake-free" white area, and one for points in the "cross-mistake" gray areas. Notably, XRM is able to allocate the two smallest groups ■■ to dedicated environments. Another notable observation is that the two middle plots appear as straight lines, indicating that the twin networks agree on their predictions. The fourth panel shows that label flipping happens almost exclusively for the two smallest groups, and stabilizes as training progresses.

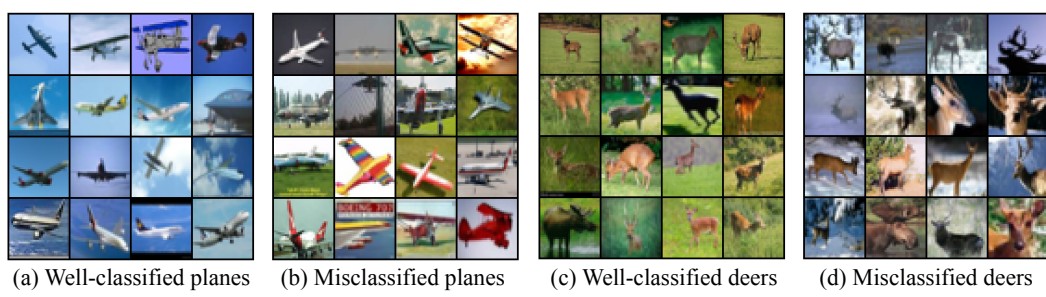

(a) Well-classified planes   (b) Misclassified planes   (c) Well-classified deers   (d) Misclassified deers

Figure 3: Randomly selected images of CIFAR-10 from groups identified by XRM. The twin networks show interesting patterns in their mistakes. Notably, well-classified examples are prototypical.

## 6  DISCUSSION

We have introduced CROSS-RISK MINIMIZATION (XRM), a simple algorithm for environment discovery. XRM provides a recipe to tune its hyper-parameters, does not require early-stopping, and can discover environments for all training and validation data—dropping the requirement for human annotations at all. More specifically, XRM trains two twin label predictors on random halves of the training data, while encouraging each twin to imitate confident held-out mistakes by their sibling. This implements an "echo-chamber" that identifies environments that differ in spurious correlation, and endow domain generalization algorithms with oracle-like performance.

We highlight two directions for future work. Firstly, how does XRM relate to the invariance principle $Y \perp E \mid \Phi(X)$? What is the interplay between revealing relevant labels $Y$ and relevant environments $E$ as to afford invariance? To our knowledge, XRM is the first environment discovery algorithm tampering with labels $Y$, thus exploring invariance—and the violation thereof—from a new angle. Because relabeling happens with a probability proportional to confidence, we expect model calibration to play a role in understanding the theoretical underpinnings of XRM, as it happened with other invariance methods (Wald et al., 2021). Overall, the theoretical analysis of XRM will call for new tools, because label-flipping steers XRM away from the Bayes-optimal predictor.

Secondly, we would like to further understand the relationship between XRM and the multifarious phenomenon of memorization. Good memorization affords invariance (*Where did I park my car?*), and therefore depends on the collection of environments deemed relevant. Bad memorization happens due to "structured over-fitting", commonly incarnated as a bad learning strategy "use a simple feature for the majority, then memorize the minority". Does XRM discover environments that promote features that benefit all examples?

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

# APPENDIX

## A   RELATED WORK (NEWLY ADDED)

The literature on domain generalization spans a decade and comprises a vast amount of works. In the review below, we survey (i) some of the major milestones in domain generalization research, (ii) advances in the sub-problem of sub-population shift, the problem attacked with our XRM proposal, (iii) the multifarious connections between domain generalization and causal inference, (iv) efforts to learn domains, sub-populations, or environments from pooled collections of training examples previous to our XRM proposal, and (v) their limitations in terms of annotation requirements and impossibility results.

(i) The first works in *domain generalization* proposed algorithms that summarize each domain as a kernel mean embedding of the respective distribution of inputs (Blanchard et al., 2011; Muandet et al., 2013); these were later extended to the realm of deep neural networks (Zhang et al., 2021). One common avenue toward domain generalization is to learn a predictor where the feature representation has the same distribution across domains (Sun and Saenko, 2016; Ganin et al., 2016). Another major strategy is to enforce learning a richer feature space (Zhang et al., 2022a), which can be done by combining the weights of multiple models with different hyper-parameter configurations (Cha et al., 2021; Rame et al., 2022; Wortsman et al., 2022; Lin et al., 2023), or biasing training to make them disagree with each other (Nam et al., 2020; Pagliardini et al., 2023; Lee et al., 2023). Learning from combinations of examples, by means of mixup (Zhang et al., 2018), is also a promising route to diminish the impact of spurious correlations (Yao et al., 2022; Giannone et al., 2022). All in all, there are multiple frameworks that evaluate dozens of domain generalization algorithms across a variety of benchmark datasets, such as DomainBed (Gulrajani and Lopez-Paz, 2020) and WILDS (Pang Wei Ko et al., 2021). We recommend the reader to consult recent surveys (Zhou et al., 2022a; Wang et al., 2021) for a taxonomy of the vast array domain generalization algorithms on offer.

(ii) *Sub-population shift* is a particular type of domain generalization problem, where environments are direct annotations of a spurious attribute, and one can assume that the test domain will be equal a subdistribution—group—of the training data. The gold-standard for addressing sub-population shifts is group distributionally robust optimization (Sagawa et al., 2019, GroupDRO). Group subsampling and reweighting schemes, albeit simple, also provide state-of-the-art accuracy (Idrissi et al., 2022). To achieve good performance, it is known that it suffices to finetune the last layer of a deep neural network with a small training set with balanced groups (Izmailov et al., 2022). The framework of SubpopBench compares twenty algorithms for sub-population shift across a dozen benchmark datasets (Yang et al., 2023).

(iii) The goal of domain generalization can be understood as finding predictors invariant across a family of relevant environments (Arjovsky et al., 2019). This establishes an intimate link between domain generalization and causality under the interventionist account, where causation is defined as invariance across interventions (Woodward, 2005). A pioneering method attacks the problem domain generalization as finding invariant causal predictors (Peters et al., 2016, ICP). The framework of invariant risk minimization (Arjovsky et al., 2019, IRM) extends ICP to deep neural networks, advocating the invariance principle of "finding a feature representation such that the optimal classifier matches across environments". Researchers have proposed multiple variants of the original IRM formulation, with notable examples being risk extrapolation (Krueger et al., 2021, vREX) and sparse risk minimization (Zhou et al., 2022b). The IRM framework has found multiple applications, with fair face recognition Ma et al. (2023) being a recent example.

(iv) The main factor limiting the application of domain generalization and sub-population shift machinery is their requirement of domain, environment, or group annotations. Unfortunately, these are resource-intensive to obtain and are limited by human annotators' biases, as the biases they identify may not align with those learned by models, and vice versa (Bell and Sagun, 2023).

Consequently, a wide array of methods has been recently proposed to estimate these annotations from pooled collections of training data. Learning from failure (Nam et al., 2020, LfF) learns a biased network, and a final network that focuses on the examples misclassified by the biased network. Environment inference for invariant learning (Creager et al., 2020, EIIL) searches for an environmental partition that violates the IRM principle. Just-train-twice (Zheran Liu

et al., 2021, JTT) trains one first network for a few iterations, and a final network to focus more on the examples from the first network. Correct and Contrast (Zhang et al., 2022b, CNC) leverages ERM failures and contrastive learning to learn a robust representation. Automatic feature reweighting (Qiu et al., 2023, AFR) learns a first network for a few iterations, and then fine-tunes the last layer to focus on mistakes. Learning to split (Bao and Barzilay, 2022, LS) and adversarial re-weighted learning (Lahoti et al., 2020, ARL) implement adversarial games to find a split of the training data inducing maximum out-of-sample error. Bias amplification (Li et al., 2023, BAM) incorporates per-example "slack variables" to absorb the fast learning of spurious correlations. No subclass left behind (Sohoni et al., 2020, GEORGE) clusters the hidden representation of a neural network to construct different environments. (Teney et al., 2021) manually identify variables to stratify pooled collections of training examples into environments.

In the field of label noise robustness, CrossSplit (Kim et al., 2023) and XRM share a similar approach by training two networks. While XRM relies on confident held-out mistakes to indicate environment annotation, CrossSplit uses them as indicators of a model's memorization of noisy labels.

(v) One important note to the environment discovery methods described above is that they still require group annotations in a validation set, used for selecting a model with good worst-group-accuracy. In the complete absence of environment annotations, learning invariant predictors is an impossible task in its full generality (Lin et al., 2022a; Tan et al., 2023; Chen et al., 2023). Because we have proposed XRM as an alternative to surmount such daunting task, the next section provides intuitions to identify success and failure cases of our method.

## B  WHEN DOES XRM WORK, AND WHEN DOES IT FAIL? (NEWLY ADDED)

The problem of learning invariant predictors in the absence of appropriate environment annotations is an impossible problem in its full generality (Lin et al., 2022a). In particular, the issue of dividing data into invariance-affording environments parallels the problem of "controlling for" the right set of variables to deconfound causal relationships (Pearl, 2009). Researchers in domain generalization and causal inference both agree that, in order to reveal the true relationship between an input $X_{\text{inv}}$ and a target $Y$ of interest, one must "control-for" a set of variables $E$ that satisfy

$$Y \perp E \mid X_{\text{inv}}. \tag{5}$$

This formula is referred to as the *invariance principle* in domain generalization (Arjovsky et al., 2019), where the variable $E$ receives the name of *environments*. In causal inference, the conditional independence statement (5) is known as *conditional exchangeability*, and the variable $E$ receives the name of *valid adjustment set* (Pearl, 2009; Hernán and Robins, 2010). As commonly suggested in clinical trials, one should control-for pre-treatment variables, while discourage to control-for post-treatment variables. For instance, stratifying based on or controlling for $E$ on data with causal structure $X_{\text{inv}} \rightarrow Y \rightarrow E$ would bias our estimate about the regression coefficient from $X_{\text{inv}}$ to $Y$.

Different causal structures can produce the same observational data Peters et al. (2017), so identifying an appropriate $E$ requires knowledge about causal structure—this is why researchers in public policy and epidemiology spend a lot of time justifying why, for the problem at hand, certain $E$ is a valid instrumental variable or adjustment set to control-for. Alas, there is no universal recipe to de-confound a relationship between two variables without admitting extra knowledge about the causal structure behind our data, and XRM is not an exception of such free lunch.

Therefore, we would expect XRM to work well in instances where the estimated environments $E$ satisfy the conditional independence statement (5), and we should anticipate trouble in those cases where the discovered environments violate (5). However, and as discussed in (Lin et al., 2022a), evaluating (5) requires knowing the invariance-inducing feature $X_{\text{inv}}$, which is the variable subject to discovery. This makes assumptions such as (5) difficult to verify in practice when collecting our environments $E$, and the best we can do is to offer some canonical examples of successes and failures, that can guide our choices of when to apply XRM.

So, we exemplify with four different versions of the ColorMNIST dataset (Arjovsky et al., 2019). All four versions instantiate a colored digit classification task, differing on whether the robust feature is "digit shape" or "digit color", and which one of these two variables bear the strongest correlation to

| | CMNIST Arjovsky et al. (2019) | InverseCMNIST Zhang et al. (2022a) | InverseMCOLOR (No citation) | MCOLOR Lin et al. (2022b) |
|---|---|---|---|---|
| training, $e = 1$ | $S \xrightarrow{0.75} Y \xrightarrow{0.80} C$ | $S \xrightarrow{0.85} Y \xrightarrow{0.70} C$ | $S \xleftarrow{0.80} Y \xleftarrow{0.75} C$ | $S \xleftarrow{0.80} Y \xleftarrow{0.85} C$ |
| training, $e = 2$ | $S \xrightarrow{0.75} Y \xrightarrow{0.90} C$ | $S \xrightarrow{0.85} Y \xrightarrow{0.80} C$ | $S \xleftarrow{0.90} Y \xleftarrow{0.75} C$ | $S \xleftarrow{0.70} Y \xleftarrow{0.85} C$ |
| training, pooled | $S \xrightarrow{0.75} Y \xrightarrow{0.85} C$ | $S \xrightarrow{0.85} Y \xrightarrow{0.75} C$ | $S \xleftarrow{0.85} Y \xleftarrow{0.75} C$ | $S \xleftarrow{0.75} Y \xleftarrow{0.85} C$ |
| testing | $S \xrightarrow{0.75} Y \xrightarrow{0.10} C$ | $S \xrightarrow{0.85} Y \xrightarrow{0.10} C$ | $S \xleftarrow{0.10} Y \xleftarrow{0.75} C$ | $S \xleftarrow{0.10} Y \xleftarrow{0.85} C$ |
| inv. feature? | complex, weak | complex, strong | simple, weak | simple, strong |
| ERM | $0.37 \pm 0.10$ | $0.67 \pm 0.02$ | $\mathbf{0.75} \pm 0.01$ | $\mathbf{0.85} \pm 0.01$ |
| XRM | $\mathbf{0.71} \pm 0.02$ | $\mathbf{0.82} \pm 0.02$ | $0.40 \pm 0.01$ | $0.57 \pm 0.01$ |
| Oracle | 0.75 | 0.85 | 0.75 | 0.85 |

Table 3: Four ColoredMNIST versions, where the environment $E$ influences digit shape $S$ and color $C$, forming our input $X = (S, C)$. We depict the causal structure for each dataset version, and the correlation between variables. The invariant feature may be the complex digit shape (CMNIST versions) *or* the simple digit color (MCOLOR versions), which in turn could bear the strongest *or* weakest correlation to the target variable—producing four versions of the ColoredMNIST problem. Note that CMNIST-MCOLOR and InverseCMNIST-InverseMCOLOR are indistinguishable from pooled training data alone. At the bottom, test accuracies of ERM, XRM+GroupDRO, and an Oracle which relies solely on the invariant feature.

the target label. Overall, we expect "digit color" to be faster (easier) to learn, leading to generalization issues when "digit shape"—more difficult and slower to learn—is the desired invariant feature.

We show in table 3 the average-test-accuracy of ERM and XRM followed by GroupDRO for the four versions of the ColoredMNIST dataset. We also show what a hypothetical oracle, relying solely on the invariant feature, would achieve. ERM performs well when the invariant feature is the simplest of the two. XRM performs well when the invariant feature is the most complex of the two. We highlight that the datasets CMNIST and MCOLOR are observationally equivalent from pooled data alone—and a similar remark follows for InverseCMNIST and InverseMCOLOR. This echoes the impossibility results of (Lin et al., 2022a), namely learning invariant predictors in the absence of environment annotations is impossible in its full generality: for instance, based on training data alone, we would never know if we are dealing with InverseCMNIST or InverseMCOLOR, and therefore we are at a loss of whether to apply ERM or XRM. Nevertheless, XRM remains an state-of-the-art solution for those problems were we would like our learning machine to ignore the fastest-to-learn feature, often being a spurious shortcut (Geirhos et al., 2020; Shah et al., 2020a; Pezeshki et al., 2021), in order to focus on more complex patterns with a higher potential for invariance.

Table 4: The average and worst test test environment accuracies for five datasets in the DOMAINBED benchmark (Gulrajani and Lopez-Paz, 2020). Three methods are compared: 1) ERM with no environment annotations, 2) CORAL with human-annotated environments, and 3) CORAL with XRM-inferred environments. The model selection is done according to the average accuracy over validation environments.

| | VLCS | | PACS | | OfficeHome | | TerraInc | | DomainNet | |
|---|---|---|---|---|---|---|---|---|---|---|
| **Method** (annotations) | Avg | Worst | Avg | Worst | Avg | Worst | Avg | Worst | Avg | Worst |
| **ERM** (None) | 77.97 | 64.85 | 83.35 | 72.55 | 65.47 | 52.25 | 47.02 | 34.60 | 31.69 | 9.30 |
| **CORAL** (Human) | 77.87 | 65.00 | 84.99 | 77.70 | 67.74 | 53.55 | 48.51 | 37.15 | 41.97 | 13.25 |
| **CORAL** (XRM) | 77.66 | 66.15 | 83.81 | 77.30 | 67.01 | 53.90 | 49.60 | 38.00 | 35.87 | 11.60 |

## C    RESULTS ON DOMAINBED (NEWLY ADDED)

Table 4 presents additional domain generalization results on the DOMAINBED benchmark (Gulrajani and Lopez-Paz, 2020). Experiments compare three settings: ERM without any environment annotations, the CORAL domain generalization algorithm (Sun and Saenko, 2016) with human-annotated environments, and CORAL with environments discovered by XRM. As a note, CORAL is the best performing single-model (non-ensembling) method in the DomainBed suite. Results suggest that the performance when using XRM-inferred annotations is comparable to that of human-annotated environments.

### C.1    FURTHER DETAILS

We adhere to the original codebase from DOMAINBED (Gulrajani and Lopez-Paz, 2020). For each dataset, we consider hold out each possible environment as the test domain, and train on the remaining environments. In table 5, we report the results for each environment when selected as the test environment. Additionally, we report the average and worst environment test accuracies.

In settings without environment annotations (ERM), we combine all training environments and then split into one training and one validation set. In those cases with annotations, whether human-annotated or discovered by XRM, each training environment is divided into as many training and validation sets as the number of environments. For each triplet of (dataset, method, test environment), we sweep over 16 different hyper-parameter combinations. We perform model selection based on the *average accuracy over the validation environments*, which is referred to as the *'training domain validation set'* in the DOMAINBED paper. The same procedure is followed for XRM, except that model selection is done with respect to the *flip rate*.

## D    EXPERIMENTAL DETAILS

For the results in table 1, we follow SubpopBench's experimental protocol (Yang et al., 2023) with a notable exception: for model selection, we forgo the 'oracle (test-set) model selection' used in SubpopBench and instead adhere to the standard practice of utilizing the validation set. Therefore, image datasets use a pretrained ResNet-50 (He et al., 2016) unless otherwise mentioned, and text datasets use a pretrained BERT (Devlin et al., 2018). All images are resized and center-cropped to $224 \times 224$ pixels, and undergo no data augmentation. We use SGD with momentum $0.9$ to learn from image datasets unless otherwise mentioned, and we employ AdamW (Loshchilov and Hutter, 2017) with default $\beta_1 = 0.9$ and $\beta_2 = 0.999$ for text benchmarks. For the ColorMNIST experiment (Arjovsky et al., 2019), we train a three-layer fully-connected network with layer sizes $[2 * 14 * 14, 300, 300, 2]$ and use ReLU as the activation function. The network is optimized using the Adam optimizer with a learning rate of $1e - 3$, and default parameters $\beta_1 = 0.9$ and $\beta_2 = 0.999$. For the experiment on CIFAR-10 (Krizhevsky et al., 2009), we train a VGG-16 model (Simonyan and Zisserman, 2014) using SGD with a learning rate of $1e - 2$ and a momentum of $0.9$ We train XRM and phase-2 algorithms for a number of iterations that allows convergence within a reasonable compute budget. These are 5,000 steps for Waterbirds and Metashift, 10,000 steps for ImageNetBG, 20,000 steps for MultiNLI, and 30,000 steps for CivilComments.

Table 5: Results for the DOMAINBED suite.

| VLCS | C | L | S | V | Avg | Worst |
|------|------|------|------|------|------|------|
| **ERM** (None) | 96.70 | 64.85 | 74.20 | 76.15 | 77.97 | 64.85 |
| **CORAL** (Human) | 97.35 | 65.00 | 72.80 | 76.35 | 77.87 | 65.00 |
| **CORAL** (XRM) | 95.55 | 66.15 | 72.45 | 76.50 | 77.66 | 66.15 |

| PACS | A | C | P | S | Avg | Worst |
|------|------|------|------|------|------|------|
| **ERM** (None) | 84.65 | 80.65 | 95.55 | 72.55 | 83.35 | 72.55 |
| **CORAL** (Human) | 84.90 | 80.75 | 96.60 | 77.70 | 84.98 | 77.70 |
| **CORAL** (XRM) | 81.90 | 77.30 | 96.90 | 79.15 | 83.81 | 77.30 |

| OfficeHome | A | C | P | R | Avg | Worst |
|------|------|------|------|------|------|------|
| **ERM** (None) | 59.50 | 52.25 | 74.15 | 76.00 | 65.47 | 52.25 |
| **CORAL** (Human) | 64.00 | 53.55 | 76.15 | 77.25 | 67.73 | 53.55 |
| **CORAL** (XRM) | 61.80 | 53.90 | 74.85 | 77.50 | 67.01 | 53.90 |

| TerraIncognita | L100 | L38 | L43 | L46 | Avg | Worst |
|------|------|------|------|------|------|------|
| **ERM** (None) | 54.80 | 42.30 | 56.40 | 34.60 | 47.02 | 34.60 |
| **CORAL** (Human) | 58.20 | 39.25 | 59.45 | 37.15 | 48.51 | 37.15 |
| **CORAL** (XRM) | 59.20 | 45.10 | 56.10 | 38.00 | 49.60 | 38.00 |

| DomainNet | clip | info | paint | quick | real | sketch | Avg | Worst |
|------|------|------|------|------|------|------|------|------|
| **ERM** (None) | 47.40 | 14.75 | 37.45 | 9.30 | 42.10 | 39.15 | 31.69 | 9.30 |
| **CORAL** (Human) | 60.05 | 20.25 | 47.90 | 13.25 | 59.95 | 50.45 | 41.97 | 13.25 |
| **CORAL** (XRM) | 50.40 | 16.80 | 42.30 | 11.60 | 50.40 | 43.70 | 35.87 | 11.60 |

### D.1 PHASE-1 XRM MODEL SELECTION

For phase-1, we run XRM with 16 random combinations of hyper-parameters, each over $k_1 = 10$ random seeds. (We repeat runs with null accuracy on one of the classes.) For each of the 16 hyper-parameter combinations, we average the number of flipped labels appearing at the last iteration (early-stopping is not necessary with XRM) across the 10 seeds. This will tell us which one of the 16 hyper-parameter combinations is best. For that combination, we average the training and validation logit matrices across the 10 random seeds. Finally, we discover environments using equation (4).

### D.2 PHASE-2 DG MODEL SELECTION

For all phase-2 domain generalization algorithms (ERM, SUBG, RWG, GroupDRO), we search over 16 random combinations of hyper-parameters. We select the hyper-parameter combination and early-stopping iteration yielding maximal worst-group-accuracy (or, in the absence of groups, worst-class-accuracy). We re-run the best hyper-parameter combination for $k_2 = 10$ random seeds to report avg/std for test worst-group-accuracies (always computed with respect to the ground-truth group annotations). The $k_1$ random seeds from phase-1 do not contribute to error bars.

### D.3 HYPER-PARAMETER SAMPLING GRIDS

| algorithm | hyper-parameter | ResNet | BERT |
|---|---|---|---|
| XRM, ERM, SUBG, RWG | learning rate
weight decay
batch size
dropout | $10^{\text{Uniform}(-4,-2)}$
$10^{\text{Uniform}(-6,-3)}$
$2^{\text{Uniform}(6,7)}$
— | $10^{\text{Uniform}(-5.5,-4)}$
$10^{\text{Uniform}(-6,-3)}$
$2^{\text{Uniform}(3,5.5)}$
$\text{Random}([0,0.1,0.5])$ |
| GroupDRO | $\eta$ | $10^{\text{Uniform}(-3,-1)}$ | $10^{\text{Uniform}(-3,-1)}$ |

## E LEARNING TO SPLIT ON WATERBIRDS

We benchmarked the official learning to split code-base https://github.com/YujiaBao/ls on the WaterBirds dataset. We assessed the method's sensitivity to two hyperparameters: the number of epochs used for early stopping (`patience` argument in the codebase) and the pre-supposed ratio of groups (based on the `ratio` argument in the code). For `patience` we swept over (2, 5, 10) with 5 being the default value. For `ratio`, we swept over (0.25, 0.5, 0.75) with 0.75 being the default value based on the paper. We found worst group performance using a fixed GroupDRO phase-2 training varied by as much as $\pm 7\%$ on Waterbirds.

## F EXPERIMENTAL RESULTS WITH ERROR BARS

| | ERM | | | GroupDRO | | | RWG | | | SUBG | | |
|---|---|---|---|---|---|---|---|---|---|---|---|---|
| | None | Human | **XRM** | None | Human | **XRM** | None | Human | **XRM** | None | Human | **XRM** |
| **Waterbirds** | 70.4 ±2.99 | 76.1 ±2.37 | 75.3 ±1.96 | 71.7 ±4.09 | 88.0 ±2.61 | 86.1 ±1.28 | 74.8 ±2.50 | 87.0 ±1.63 | 84.5 ±1.53 | 73.0 ±2.75 | 86.7 ±1.00 | 76.3 ±8.41 |
| **CelebA** | 62.7 ±2.73 | 71.9 ±3.48 | 67.6 ±3.48 | 68.8 ±1.29 | 89.1 ±1.67 | 89.8 ±1.39 | 70.7 ±1.32 | 89.5 ±1.45 | 88.0 ±2.56 | 68.5 ±2.13 | 87.1 ±2.70 | 87.5 ±2.54 |
| **MultiNLI** | 54.8 ±4.04 | 65.3 ±3.02 | 65.8 ±3.41 | 69.7 ±2.65 | 76.1 ±1.29 | 74.3 ±1.88 | 69.3 ±1.77 | 71.1 ±1.60 | 73.3 ±1.56 | 53.7 ±2.97 | 72.8 ±0.66 | 71.3 ±1.58 |
| **CivilComments** | 55.1 ±3.46 | 59.7 ±5.77 | 61.4 ±4.48 | 59.3 ±2.05 | 69.3 ±2.32 | 71.3 ±1.35 | 54.0 ±4.58 | 65.8 ±6.30 | 73.4 ±0.93 | 58.7 ±2.56 | 64.0 ±7.63 | 72.9 ±1.12 |
| **ColorMNIST** | 10.1 ±0.51 | 10.1 ±2.40 | 26.9 ±2.27 | 10.0 ±0.51 | 10.1 ±2.37 | 70.5 ±0.98 | 10.1 ±0.51 | 10.2 ±1.85 | 70.2 ±1.00 | 10.1 ±0.51 | 10.0 ±2.21 | 70.8 ±1.09 |
| **MetaShift** | 70.8 ±2.99 | 67.7 ±4.62 | 71.2 ±3.95 | 70.8 ±3.91 | 73.7 ±4.42 | 71.2 ±4.81 | 66.5 ±4.55 | 70.8 ±4.45 | 69.7 ±4.86 | 70.0 ±3.38 | 73.5 ±3.49 | 69.2 ±5.58 |
| **ImagenetBG** | 75.6 ±3.04 | 76.9 ±1.89 | 76.9 ±1.93 | 73.0 ±3.43 | 76.1 ±1.40 | 75.9 ±1.69 | 76.9 ±2.42 | 76.5 ±2.65 | 77.0 ±2.66 | 75.0 ±3.55 | 75.0 ±3.55 | 76.4 ±1.79 |

# G  XRM IN PYTORCH

```python
1  import torch
2
3  def balanced_cross_entropy(p, y):
4      losses = torch.nn.functional.cross_entropy(p, y, reduction="none")
5      return sum([losses[y == yi].mean() for yi in y.unique()])
6
7  def xrm(x_tr, y_tr, x_va, y_va, lr=1e-2, max_iters=1000):
8      # init twins, assign examples, and calibrate (Section 4.1)
9      nc = len(y_tr.unique())
10     net_a = torch.nn.Linear(x_tr.size(1), nc)
11     net_b = torch.nn.Linear(x_tr.size(1), nc)
12     ind_a = torch.zeros(len(x_tr), 1).bernoulli_(0.5).long()
13
14     # Platt temperature scaling
15     temp_a = torch.nn.Parameter(torch.ones(1, nc))
16     temp_b = torch.nn.Parameter(torch.ones(1, nc))
17     logits_a = net_a(x_tr).detach()
18     logits_b = net_b(x_tr).detach()
19     cal = torch.optim.SGD([temp_a, temp_b], lr)
20
21     for iteration in range(max_iters):
22         logits = logits_a / temp_a * ind_a + logits_b / temp_b * (1 - ind_a)
23         cal.zero_grad()
24         balanced_cross_entropy(logits, y_tr).backward()
25         cal.step()
26
27     net_a.weight.data.div_(temp_a.t().detach())
28     net_b.weight.data.div_(temp_b.t().detach())
29
30     # training (Section 4.2)
31     opt = torch.optim.SGD(
32         list(net_a.parameters()) + list(net_b.parameters()), lr)
33
34     for iteration in range(max_iters):
35         pred_a, pred_b = net_a(x_tr), net_b(x_tr)
36         pred_hi = pred_a * ind_a + pred_b * (1 - ind_a)
37         pred_ho = pred_a * (1 - ind_a) + pred_b * ind_a
38
39         opt.zero_grad()
40         balanced_cross_entropy(pred_hi, y_tr).backward()
41         opt.step()
42
43         # label flipping, useful for model selection (Section 4.3)
44         p_ho, y_ho = pred_ho.softmax(dim=1).detach().max(1)
45         is_flip = torch.bernoulli((p_ho - 1 / nc) * nc / (nc - 1)).long()
46         y_tr = is_flip * y_ho + (1 - is_flip) * y_tr
47
48     # environment discovery (Section 4.4)
49     cm = lambda x, y: torch.logical_or(
50         net_a(x).argmax(1).ne(y),
51         net_b(x).argmax(1).ne(y)).long().detach()
52
53     return cm(x_tr, y_tr), cm(x_va, y_va)
```

The code above may be helpful to clarify our exposition in the main text. For an end-to-end example running linear XRM and GroupDRO, see: https://pastebin.com/0w6gsxQw.

