# OpenReview forum: "Discovering Environments with XRM"
_ICLR.cc/2024/Conference — Submitted to ICLR 2024_

### Official Review · Reviewer_KDkA · 2023-10-13

**Soundness:** 2 fair
**Presentation:** 2 fair
**Contribution:** 2 fair
**Rating:** 3
**Confidence:** 2

**Summary:**

The paper addresses OOD generalization by discovering latent environments (partitions) of the training data that are beneficial when used subsequently with standard methods (GroupDRO, reweighting, or resampling to equalize groups during training). The method proceeds by training a pair of models (details discussed below).

**Strengths:**

- Thorough evaluation on multiple standard datasets.

- Good empirical results.

**Weaknesses:**

W1. If I understand correctly, the method seems to rely on the fact that misclassified examples are such because they do not contain a "spurious correlation" that a model would learn by default. The twin training serves to reinforce the tendency of one of the trained models to capture this spurious correlation. If this is indeed the case, then the overall methods seems to depend on the (common) heuristic that models learn spurious correlations by default (a.k.a. shortcut learning). I think this is the same heuristic that is used in the existing methods criticised in Section 3. Critically, this heuristic relies on the fact that we know that the chosen architecture/training set lead to learnin undesirable spurious correlations by default. What if one applies the method to a situation where a perfectly-fine, "robust" model is learned by default? I'm guessing that the method would then be detrimental.

I'm not suggesting that we should be able to do better without additional knowledge (in fact [1] seems to show it's not possible) but the authors here do claim to do so, hence the need to point out this possible limitation (see also W5).

If my understanding of the method is correct, the method is also very similar to the following.
- Works in the debiasing literature (e.g. LfF) that train a pair of models that respectively rely/do not rely on spurious features. These works are discussed in Sect. 3, and I do understand that they rely to some extent to the tuning of model capacity to ensure that it captures the spurious feature, but I am not convinced that the proposed parallel training (which seem to be the essential difference) leads to the discovery of something fundamentally different.
- Works in the "model diversity" literature that train a pair of models that differ in their predictions [5,6]. These also proceed to train models in parallel, in a was that seems conceptually very similar to the step 1 proposed here (implementation details aside).

--------

W2. The proposed method only partitions the data into 2 "environments". I don't think this is really in line with the literature on DG (with which this work is supposed to connect) that are mostly based on invariance learning and need a large number of training environments (e.g. IRM). This work therefore seems much more related to the simpler setting of "debiasing" methods (e.g. LfF) that aim at removing the reliance of a model from one precise biased feature.

The methods used for the second phase are indeed simple baselines for debiasing, and not really DG methods. These are very strong baselines in these settings (and with the datasets considered), but I'm not sure this is what the reader would expect given all the mentions about DG.

--------

W3. Absence of a comprehensive review of related work. Some directly-related methods are correctly cited/discussed throughout the paper, but there are other connected areas that are not really discussed (examples below).

[1,2] discuss conditions under which environment discovery is possible. I think the theoretical statements in [1] are particularly important to discuss (I am not sure how the proposed method overcomes the impossibility stated in that paper; see also W5).
[3] was an early method that also proposed to "unshuffle" data (a term used in this paper) by simply clustering the data. Looking at the visualizations of discovered "environments" in Fig. 3, one wonders if these could also be discovered with such as simple clustering baseline.
[4] is another recent method that also seems to claim discovering partitions in the data (I suspect it has similar flaws to those discussed in the paper; it has appeared at ICCV 2023 after the ICLR deadline so it's totally fine to dismiss it though).

[1] [ZIN When and How to Learn Invariance Without Environment Partition](https://arxiv.org/abs/2203.05818)

[2] [Provably Invariance Learning without Domain Information](https://proceedings.mlr.press/v202/tan23b/tan23b.pdf)

[3] [Unshuffling data for improved generalization](https://arxiv.org/abs/2002.11894)

[4] [Invariant Feature Regularization for Fair Face Recognition](https://openaccess.thecvf.com/content/ICCV2023/papers/Ma_Invariant_Feature_Regularization_for_Fair_Face_Recognition_ICCV_2023_paper.pdf)

[5] [Agree to Disagree: Diversity through Disagreement for Better Transferability](https://arxiv.org/abs/2202.04414)

[6] [Diversify and Disambiguate: Learning From Underspecified Data](https://arxiv.org/abs/2202.03418)

--------

W4: No discussion or empirical exploration of the limitations of the methods. No precise statement of the assumptions on which the method relies.

--------

Minor comments (no need to comment in the rebuttal; these do not affect my rating of the paper)

- W-minor 1. The writing style is unusual for a technical paper. There are many verbose statements, emotional words, exclamation marks, etc. This is actually a great writing style in other circumstances, but it does not maximize the clarity and efficiency of communication. This does not directly affect my rating of the paper, but it made the reading more tedious. I would suggest using a more concise style and neutral tone for the benefit of the readers.

- W-minor 2. The existing methods for environment discovery based on 2 phases is described twice in sections 1 and 3. It could be clearer to merge these.
Section 3 is a mix of review/background material/motivation/related work. It's not bad at all in its contents, but it could be easier for the readers to stick with common sections like "related work", "background", etc.

- W-minor 3. Note that the initial premise stated in the very first sentence of the abstract is not really correct (although it does not really affect the rest of the paper):
"Successful out-of-distribution generalization requires environment annotations (...) therefore (...) we must develop algorithms to automatically discover environments"
Using multiple training environments/domains is only one approach to improve OOD generalization.

**Questions:**

Please comment on W1-W4 above.

To summarize, the main reasons for my negative rating are the absence of precise statements about limitations/assumptions of the method, and the missing discussion of links with the existing literature. Therefore, I am not sure this is really a work about DG (but rather the simpler setting of single-bias), and the core of the method may be very similar to existing work [5,6] (although presented in very different terms).

--------

In the spirit of constructive feedback, I would suggest that theses issues are fixable (in a future version) with:

(1) a proper review of the existing work, how/if it relates to this work (e.g. what is the connection with invariance learning? how do the many-environment methods related to this one? how to understand the claims made here in relation to the impossibility theorem in [1] mentioned below?)

(2) a better discussion why/how the proposed method work. The current text is mostly hand waving. Even if a complete theory is out of reach, perhaps a concrete example could help (conceptual, or with a toy example).

---

> ### Author Response · Authors · 2023-11-21
> **Addressing Review Feedback**
>
> Thank you so much for your valuable feedback.
>
> > What if one applies the method to a situation where a perfectly-fine, "robust" model is learned by default?
>
> Appendix B now contains a tour on four versions of the ColoredMNIST dataset, providing intuition about those cases where XRM works, and those instances where it may fail. As you suggest, XRM is very helpful when the spurious correlation is learned first, whereas ERM is a better alternative in those cases where the invariant correlation is learned first. In sum, we recommend running both ERM and XRM when the spurious correlation type is unknown.
>
> > [Comparison to LfF]
>
> We now clarify (in blue). The crucial ingredient in XRM is the mechanism of label-flipping, which allows us to (i) run our environment discovery method safely until convergence, and (ii) count label flips as a model selection criteria. This is a major advancement with respect to previous methods, which (i) required surgical early stopping, and (ii) didn't provide a model selection criteria. Consequently, previous work resorted to a validation set with environment annotations, the very information subject to discovery.
>
> > [Comparison to model diversity]
>
> We now clarify (in blue). Once again, the difference is in the label-flipping mechanism, which implements an "echo chamber" that biases the twins to increasingly rely on spurious correlations, and *agree* on their predictions. This is in fact the opposite of what is pursued in the model diversity literature—training multiple accurate models that differ in their predictions.
>
> > The proposed method only partitions the data into 2 "environments"
>
> Whether two environments suffice—when constructed appropriately—is in fact a fascinating question that we have pondered about for a while. We do not believe that XRM is identifying "one spurious correlation" but, more generally, "the direction of spuriousness”. We lack the theory we would love to have here, but we suspect that an Oracle with knowledge about what the test domain will be would be able to split the training data as to maximally reveal the train-test discrepancy. Thank you for pointing this out, we now discuss this stimulating topic (in blue).
>
> > I'm not sure this is what the reader would expect given all the mentions about DG
>
> We are currently conducting experiments on DomainBed, a key benchmark in domain generalization. We will include the findings in Appendix C in coming hours.
>
> > Absence of a comprehensive review of related work
>
> Appendix A now contains an exhaustive literature review on domain generalization, subpopulation shift, and their relations to invariance and causal inference. We survey methods for environment discovery, as well as the relevant impossibility results and other interesting pieces of work you kindly mention.
>
> > No discussion or empirical exploration of the limitations of the methods
>
> Again, Appendix B now presents a ColoredMNIST dataset analysis, showing XRM's effectiveness and limitations in certain scenarios and recommending a combined use of ERM and XRM when the correlation type is unclear.
>
> ---
>
> We've done our best to address your concerns. If you need more clarification, please let us know. If you find our response satisfactory, we would appreciate it if you could consider increasing your score.

---

> > ### Comment · Reviewer_KDkA · 2023-11-22
> > **Response to the authors**
> >
> > I appreciate the efforts of the authors in patching the issues pointed out in the review. However I still think that the whole story about environment discovery does not match with the actual contributions centered on a "debiasing" (2-environment setting). Therefore I still find the paper misleading and a disservice for the readers in its current form. The relation with the existing literature should be more precisely and clearly laid out in the text, not just as an afterthought in the appendix.
> >
> > Similarly the discussion about the impossibility of solving the problem being addressed in its generality, which the authors added as Appendix B, should also be part of the main story (at least in a short form, not just as a reference to Appendix B). The bottom line seems simply that the method should be tried empirically to check whether its heuristics are appropriate for the task. This is fine since this seems to be a fundamental limitation, but it doesn't match the story and claims in the main text of a method that comes with batteries included.
> >
> > This is still a clear reject for me but I look forward to seeing another version of this paper submitted at a future venue.

---

> > > ### Author Response · Authors · 2023-11-23
> > >
> > > Thank you for your reply. We wish to find some middle ground with our answers below.
> > >
> > > > centered on a "debiasing" (2-environment setting)
> > >
> > > Some prominent works on the issue of learning from multiple environments, such as the original IRM paper, make extensive use of the 2-environment setting. Other examples include Environment Inference for Invariant Learning (EIIL), Learning to Split (LS), and most other environment discovery methods based on classification error (e.g., JTT, LfF).
> > >
> > > As now discussed in our draft, we could stratify continuous loss values into a predefined number of environments. Also, in the case of having C labels, vanilla XRM stratifies data into 2*C groups.
> > >
> > > We also encourage the reviewer to take a look at our DomainBed results, which we are very excited about, and justifies the extension of XRM to commonplace domain generalization settings.
> > >
> > > Finally, we believe that better understanding if "two environments is all you need", if those environments best reflect the expected train-test distribution shift, is an interesting direction for future work.
> > >
> > > > literature should be more precisely and clearly laid out in the text [...] Appendix B, should also be part of the main story
> > >
> > > We commit to doing so in the camera ready version. Because of space constraints, this is not a change that we can make in a short period of rebuttal time.
> > >
> > > > the method should be tried empirically to check whether its heuristics are appropriate for the task
> > >
> > > Due to no-free-lunch theorems, no learning method can work across the board, and must necessarily rest on some data assumptions, which are in many cases difficult to verify. Our new Appendix B (soon in the main text) provides the main intuition: XRM is effective when the spurious features are simpler than the invariant patterns of interest. In any case, to cover more ground, we recommend practitioners to add ERM runs to their sweeps.
> > >
> > > > claims in the main text of a method that comes with batteries included
> > >
> > > We would like to believe that the model selection criteria included in XRM is a significant step forward in the literature of environment/group inference. Previous methods, because of the lack of such a model selection criterion, had to resort to human annotations in a validation set.

---

### Official Review · Reviewer_f39H · 2023-10-15

**Soundness:** 2 fair
**Presentation:** 3 good
**Contribution:** 3 good
**Rating:** 6
**Confidence:** 4

**Summary:**

The paper introduces CROSS-RISK MINIMIZATION (XRM), a method for achieving robust out-of-distribution generalization without relying on resource-intensive environment annotations. By training twin networks to imitate confident mistakes made by each other, XRM enables automatic discovery of relevant environments for training and validation data. The proposed approach addresses the challenge of hyper-parameter tuning and achieves oracle worst-group accuracy, offering a promising solution for broad generalization in AI systems.

**Strengths:**

* The problem of learning OOD robust model without manual domain partition is a very important task, which might has great impact on real-world applications.
* The proposed method has a clear advantage over existing methods such as EIIL and JTT, that they does not need to explicitly tune the hyperparameter for early stopping. Since hyper-parameter tuning is a crucial challenge, the proposed method would be of interest to many.
* The empirical performance is strong.

**Weaknesses:**

I have several concerns as follows:

1. The paper should provide a clear discussion on the identifiability challenges presented in [1], which demonstrate that learning invariance without domain partition can be generally impossible. It is crucial to address the need for imposing inductive bias, additional assumptions, conditions, or auxiliary information to ensure the effectiveness of the proposed method. A thorough exploration of these aspects would enhance the paper's theoretical foundation and its practical applicability.

2. According to [2, 3], spurious features are defined as any nodes in the causal graph other than the direct causes of the label. However, I have concerns about the evaluations conducted on datasets like waterbird, which explicitly contain only one dominating spurious feature. These datasets may not fully reflect the implications of the proposed methods on more realistic and high-dimensional datasets such as ImageNet variants. Moreover, the paper relies on the assumption that Empirical Risk Minimization (ERM) learns spurious features first, but it may not hold true for all types of spurious features as discussed in [2, 3]. It would be valuable to address these concerns and provide further insights into the generalizability of the method to diverse real-world datasets.

3. If a large number of spurious features are present, [1] demonstrates that there are necessary and sufficient conditions for learning invariance without explicit domain partitions, which can be quite restrictive. I have concerns about whether the proposed two-stage method can effectively address this problem given the limitations imposed by these conditions. It would be valuable for the authors to discuss how their method overcomes or accommodates these restrictions and whether it can achieve satisfactory results in scenarios with a significant number of spurious features.

4.  Several studies [5, 6] have highlighted the challenges associated with learning invariance in the presence of many spurious features. In a recent paper, [4] discovered that when dealing with a large number of spurious features, each ERM model tends to learn a subset of these features. [4] further demonstrates that rather than exclusively focusing on learning invariant features, it is beneficial for OOD performance to diversify the learned spurious features (referred to as spurious feature diversification). Spurious feature diversification is shown to explain the effectiveness of empirically strong methods like SWAD and Model soup. It would be valuable to investigate whether the proposed method (XRM) can enhance spurious feature diversification and demonstrate effective performance on a broader range of real-world datasets, such as PACS, OfficeHome, DomainNet, or ImageNet variants.

Correct me if I was wrong. I would increase the score if (part of) my concerns were addressed.

[1] Yong Lin et.al., ZIN: When and how to learn invariance without domain partition.

[2] Martin Arjovsky et.al., Invariant Risk Minimization

[3] Jonas Peters, et.al.,. Causal inference using invariant prediction: identification and confidence intervals

[4] Yong Lin et.al., Spurious Feature Diversification Improves Out-of-distribution Generalization

[5] Ishaan Gulrajani et.al., In Search of Lost Domain Generalization

[6] Elan Rosenfeld et.al., The Risks of Invariant Risk Minimization

[7] Junbum Cha et.al., SWAD: Domain Generalization by Seeking Flat Minima

[8] Mitchell Wortsman et.al., Model soups: averaging weights of multiple fine-tuned models improves accuracy without increasing inference time

**Questions:**

See weakness.

---

> ### Author Response · Authors · 2023-11-21
> **Addressing Review Feedback**
>
> Thank you so much for your valuable feedback.
>
> > The paper should provide a clear discussion on the identifiability challenges presented in [ZIN]
>
> Thank you for the very interesting reference [ZIN]. Appendix B now contains a tour on four versions of the ColoredMNIST dataset, providing intuition about those cases where XRM works, and those instances where it may fail. We frame this analysis within the impossibility results described in the ZIN paper.
>
> > I have concerns about the evaluations conducted on datasets like waterbirds
>
> Our main evaluation was carried out on seven datasets, where we would argue that Metashift and ImageNetBG are "ImageNet-like". We added as many datasets as we could within our constraints in an attempt to cover a wide range of spurious correlations.
>
> > discuss how their method overcomes or accommodates these [ZIN] restrictions
>
> Appendix B now contains a tour on four versions of the ColoredMNIST dataset, providing intuition about those cases where XRM works, and those instances where it may fail. As you suggest, XRM is very helpful when the spurious correlation is learned first, whereas ERM is a better alternative in those cases where the invariant correlation is learned first. In sum, we recommend running both ERM and XRM when the spurious correlation type is unknown.
>
> > investigate whether the proposed method (XRM) can enhance spurious feature diversification and demonstrate effective performance on [...] PACS, OfficeHome, DomainNet, [...]
>
> Thank you for pointing us to feature diversification, we now discuss it in our manuscript (in blue).
> Additionally, per your suggestion, we are currently conducting experiments on DomainBed, and we will include the findings in Appendix C in coming hours.
>
> ---
>
> We've done our best to address your concerns. If you need more clarification, please let us know. If you find our response satisfactory, we would appreciate it if you could consider increasing your score.

---

> > ### Comment · Reviewer_f39H · 2023-11-22
> >
> > Thank you for the effort in addressing the concerns. I think Appendix B is important and I would appreciate it if the authors could move it to the main part in the final version. I raise my score to 6 to acknowledge this.
> >
> > As for the concerns on multiple spurious features, I think it would be more convincing to conduct experiments on MultiColorMNIST (or its variants) in [1] to see how XRM performs.
> >
> > [1] Yong Lin et.al., Spurious Feature Diversification Improves Out-of-distribution Generalization

---

> > > ### Author Response · Authors · 2023-11-23
> > > **MultiColorMNIST**
> > >
> > > We appreciate your input and have taken it into account by conducting experiments with the MultiColorMNIST dataset [1].
> > >
> > > This dataset contains multiple spurious correlations, where each image displays a central grayscale digit at the center and 32 color patches all around. In the training and validation sets, these patches are highly correlated with the labels (0.9), but in the test set, this correlation is inverted. The following table compares the worst-group test accuracy of ERM against XRM+GroupDRO (inferred group labels plus GroupDRO) across five test sets with varying levels of inverse color correlation (p):
> > > | p            | 0.90   | 0.85   | 0.80   | 0.75   | 0.70   |
> > > |--------------|--------|--------|--------|--------|--------|
> > > | ERM          | 0.26   | 0.40   | 0.55   | 0.68   | 0.79   |
> > > | XRM+GroupDRO | 0.29   | 0.44   | 0.56   | 0.72   | 0.80   |
> > >
> > >
> > > Despite the difficulty of the problem, XRM is still able to identify a set of environments for GroupDRO to improve generalization. Here, XRM divides the dataset into 20 groups (10 classes x 2[mistakes or not]). Our hypothesis is that if an example is a mistake when held out, it has "at least" one spurious correlation. That is why we expect XRM to be helpful even in the presence of multiple spurious correlations.  However, it's important to note that when multiple spurious correlations are present, their interactions become complicated, and more advanced methods may be required for real-world datasets with such scenarios.
> > >
> > > We will make sure to include these results along with a discussion on this matter in the final version. Please let us know if this addresses your concerns.
> > >
> > > Thank you again!
> > >
> > > ---
> > > [1] Yong Lin et.al., Spurious Feature Diversification Improves Out-of-distribution Generalization

---

> ### Comment · Reviewer_f39H · 2023-11-23
>
> Thank you for the additional experiments.
>
> I would be very interested in seeing future works  which is designed for multi and even high-dimensional spurious features. The spurious features are natrually high dimensional.

---

### Official Review · Reviewer_fj8c · 2023-10-29

**Soundness:** 2 fair
**Presentation:** 3 good
**Contribution:** 2 fair
**Rating:** 5
**Confidence:** 4

**Summary:**

This paper addresses the challenge of achieving robust out-of-distribution generalization without relying on resource-intensive environment annotations. The authors propose Cross-Risk Minimization (XRM), a novel approach that trains twin networks to learn from random halves of the training data while imitating confident mistakes made by their counterparts. XRM enables automatic discovery of environments for both training and validation data. The authors demonstrate the effectiveness of XRM by building domain generalization algorithms based on the discovered environments, achieving oracle worst-group-accuracy.

**Strengths:**

1. The paper is well-organized and easy to understand.
2. This paper addresses a crucial challenge in Domain Generalization (DG) tasks, which is the data-splitting process without relying on human annotations.
3. The authors provide strong empirical evidence through extensive experiments to substantiate the effectiveness of their proposed XRM method.

**Weaknesses:**

1. The paper's claims may be slightly overstated. While the focus on subpopulation shift in distribution shift is indeed important, it might be more appropriate to avoid claiming to solve a long-standing problem in out-of-distribution generalization without further empirical studies on widely recognized DG benchmarks such as DomainBed and Wilds. These additional experiments could provide more convincing evidence of the proposed approach's effectiveness.
2. The paper lacks a comprehensive discussion of important related works concerning data splitting strategies for improved DG performance and subpopulation shift, such as references [1], [2], [3] and [4]. Notably, in [1], the authors have theoretically demonstrated the challenges of learning the invariant correlation between samples and labels in the absence of prior information. Including these relevant works would enhance the paper's literature review and contextualize the proposed approach.

Typo:

In the first sentence of the paragraph above section 4.2, it appears that the authors have inadvertently added a redundant "we.”

[1] ZIN: When and How to Learn Invariance Without Environment Partition?
[2] Provably Invariant Learning without Domain Information.
[3] Rethinking Invariant Graph Representation Learning without Environment Partitions.
[4] Just Mix Once: Worst-group Generalization by Group Interpolation.

**Questions:**

1. As highlighted in [1], it is crucial to understand the specific scenarios where XRM is expected to be effective. Therefore, it would be beneficial for the authors to provide further insights into the data distribution settings in which XRM is likely to perform well. Alternatively, the authors could explore providing theoretical guarantees to enhance the understanding of XRM's strengths and limitations.
2. The observation that XRM outperforms Human-annotation methods is intriguing and warrants further explanation.

---

> ### Author Response · Authors · 2023-11-21
> **Addressing Review Feedback**
>
> Thank you so much for your valuable feedback.
>
> > The paper's claims may be slightly overstated [...] without further empirical studies on widely recognized DG benchmarks such as DomainBed
>
> In line with your suggestion, we are currently conducting experiments on DomainBed. We will share the findings in Appendix C in coming hours.
>
> > The paper lacks a comprehensive discussion of important related works.
>
> Appendix A now contains an exhaustive literature review on domain generalization, subpopulation shift, and their relations to invariance and causal inference. We also survey methods for environment discovery, as well as the relevant impossibility results you kindly mentioned.
>
> > It is crucial to understand the specific scenarios where XRM is expected to be effective
>
> Appendix B now contains a tour on four versions of the ColoredMNIST dataset, providing intuition about those cases where XRM works, and those instances where it may fail.
>
> > The observation that XRM outperforms Human-annotation methods [...] warrants further explanation
>
> Our experiments on Waterbirds, CelebA, MultiNLI, and CivilComments show that environment annotations discovered by XRM afford the same or better state-of-the-art test worst-group-accuracy than those environment annotations included in the original datasets by humans. Our hypothesis is that the biases identified by humans may not always align with the biases that models learn, and vice versa [1].
>
> ---
>
> We've done our best to address your concerns. If you need more clarification, please let us know. If you find our response satisfactory, we would appreciate it if you could consider increasing your score.
>
> [1] Bell, S.J. and Sagun, L., 2023, June. Simplicity Bias Leads to Amplified Performance Disparities.

---

### Author Response · Authors · 2023-11-23
**Change log of the submission**

Dear Reviewers,

We greatly appreciate your feedback and have diligently worked to enhance our manuscript.
Below is a short summary of the significant revisions we've made:

1. **Experiments on DomainBed**:
We have conducted additional experiments on the widely recognized DomainBed benchmark. The results of these experiments can be found in Appendix C, Tables 4 and 5. Table 4 (pasted below) shows that XRM achieves comparable, or for some even better, worst-environment accuracy than human-annotated environments.
| Method (annotations) | VLCS (Avg / Worst) | PACS (Avg / Worst) | OfficeHome (Avg / Worst) | TerraInc (Avg / Worst) | DomainNet (Avg / Worst) |
| --- | --- | --- | --- | --- | --- |
| ERM (None) | 77.97 / 64.85 | 83.35 / 72.55 | 65.47 / 52.25 | 47.02 / 34.60 | 31.69 / 9.30 |
| CORAL (Human) | 77.87 / 65.00 | 84.99 / 77.70 | 67.74 / 53.55 | 48.51 / 37.15 | 41.97 / 13.25 |
| CORAL (XRM) | 77.66 / 66.15 | 83.81 / 77.30 | 67.01 / 53.90 | 49.60 / 38.00 | 35.87 / 11.60 |

2. **Expanded Literature Review**: In response to the concerns about the depth of our literature review, we have added a new section in Appendix A with an *exhaustive* survey covering domain generalization, subpopulation shift, invariance, causal inference, environment discovery, and relevant impossibility results. This comprehensive review should address the gaps previously noted.

3. **Analysis of success and failure cases of XRM on variations of ColoredMNIST**: In Appendix B, we present an in-depth analysis of four ColoredMNIST dataset variants to address the identifiability challenges discussed in the [ZIN] paper, offering context for the assumptions underlying XRM.

4. **Enhanced Discussions**: Throughout our manuscript, we have added additional explanations (in blue) to ease the understanding of our work. These clarifications include further insights into XRM methodology, the contribution of label-flipping, the differences from model diversity approaches, and scenarios involving multiple spurious correlations.

We've made revisions to address your concerns and improve our work. Your feedback has been invaluable, and we hope these updates have addressed your concerns.

---

### Meta-Review · Area_Chair_SSVb · 2023-12-10

**Metareview:**

This paper addresses the problem of domain generalization without domain annotations. This line of research is fraught with obvious identification problems, many of which have been pointed out clearly in prior art. "OOD generalization" is not in general possible, and what  are the relevant component environments/populations and what consitute "spurious" features are identifiable from a single dataset. So the problem is at once ill-posed and not supported by a compelling use case or empirical test-bed. The paper proposes a technique for data splitting and a heuristic procedure for "imitating confident held-out mistakes". The paper received thoughtful reviews. All reviews pointed out the fundamental identification questions which are not addressed satisfyingly in the presented paper. The authors engaged in a discussion with reviewers. Unfortunately, far the most comprehensive review came from KDkA who states, reasonably, "I am not sure this is really a work about DG (but rather the simpler setting of single-bias), and the core of the method may be very similar to existing work". Therefore I am recommending rejection and suggest that the authors use this as an opportunity to improve and possibly to reposition the work.

**Justification For Why Not Higher Score:**

Unclear setting, poorly positioned contribution.

**Justification For Why Not Lower Score:**

N/A

---

### Decision · Program_Chairs · 2024-01-16

Reject